# ADVERSARIAL IMITATION LEARNING VIA BOOSTING

**Jonathan D. Chang**
Department of Computer Science
Cornell University
jdc396@cornell.edu

**Dhruv Sreenivas** *
Department of Computer Science
Cornell University
ds844@cornell.edu

**Yingbing Huang** *
Department of Electrical and Computer Engineering
University of Illinois Urbana-Champaign
yh21@illinois.edu

**Kianté Brantley**
Department of Computer Science
Cornell University
kdb82@cornell.edu

**Wen Sun**
Department of Computer Science
Cornell University
ws455@cornell.edu

## ABSTRACT

Adversarial imitation learning (AIL) has stood out as a dominant framework across various imitation learning (IL) applications, with Discriminator Actor Critic (`DAC`) (Kostrikov et al., 2019) demonstrating the effectiveness of off-policy learning algorithms in improving sample efficiency and scalability to higher-dimensional observations. Despite `DAC`'s empirical success, the original AIL objective is on-policy and `DAC`'s ad-hoc application of off-policy training does not guarantee successful imitation (Kostrikov et al., 2019; 2020). Follow-up work such as `ValueDICE` (Kostrikov et al., 2020) tackles this issue by deriving a fully off-policy AIL objective. Instead in this work, we develop a novel and principled AIL algorithm via the framework of *boosting*. Like boosting, our new algorithm, `AILBoost`, maintains an ensemble of *properly weighted* weak learners (i.e., policies) and trains a discriminator that witnesses the maximum discrepancy between the distributions of the ensemble and the expert policy. We maintain a weighted replay buffer to represent the state-action distribution induced by the ensemble, allowing us to train discriminators using the entire data collected so far. In the weighted replay buffer, the contribution of the data from older policies are properly discounted with the weight computed based on the boosting framework. Empirically, we evaluate our algorithm on both controller state-based and pixel-based environments from the DeepMind Control Suite. `AILBoost` outperforms `DAC` on both types of environments, demonstrating the benefit of properly weighting replay buffer data for off-policy training. On state-based environments, `AILBoost` outperforms `ValueDICE` and `IQ-Learn`(Garg et al., 2021), achieving competitive performance with as little as one expert trajectory.

## 1 INTRODUCTION

Imitation learning (IL) is a promising paradigm for learning general policies without rewards from demonstration data, achieving remarkable success in autonomous driving (Bronstein et al., 2022; Pomerleau, 1988), video games (Baker et al., 2022; Shah et al., 2022) and graphics (Peng et al., 2021). Adversarial Imitation Learning (AIL) is an incredibly successful approach for imitation learning (Ho & Ermon, 2016; Fu et al., 2018; Kostrikov et al., 2019; Ke et al., 2020). These methods cast IL as a distribution matching problem whereby the learning agent minimizes the divergence between the expert demonstrator's distribution and the state-action distribution induced by the agent. First

---

*Equal Contribution

introduced by (Ho & Ermon, 2016), this divergence minimization can be achieved in an iterative procedure reminiscent of GAN algorithms (Goodfellow et al., 2014) with our learned reward function and policy being the discriminator and generator respectively.

Originally, a limitation of many AIL methods was that they were *on-policy*. That is, for on-policy AIL methods like `GAIL` (Ho & Ermon, 2016) and `AIRL` (Fu et al., 2018), the algorithm would draw fresh samples from the current policy in every iteration for the distribution matching process while discarding all old samples, rendering the sample complexity of these algorithms to be prohibitively large in many applications. Follow-up works (Kostrikov et al., 2019; Sasaki et al., 2019) attempt to relax the *on-policy* requirement by creating *off-policy* methods that utilize the entire history of observed data during the learning process. This history is often represented by a replay buffer and methods such as Discriminator Actor Critic (`DAC`) show large improvements in scalability and sample complexity over their on-policy counterparts. However, these methods modify the distribution matching objective as a divergence minimization between the replay buffer's and the expert's distribution, losing the guarantee of matching the expert's behavior.

Algorithms like `ValueDICE` (Kostrikov et al., 2020) address this problem by deriving a new formulation of the AIL divergence minimization objective to be entirely off-policy. `ValueDICE`, however, in principle relies on the environments to have deterministic dynamics.[1] In this work, we consider a new perspective towards making AIL off-policy. We present a new principled off-policy AIL algorithm, `AILBoost`, via the gradient boosting framework (Mason et al., 1999). `AILBoost` maintains an ensemble of properly weighted weak learners or policies as well as a weighted replay buffer to represent the state-action distribution induced by our ensemble. Our distribution matching objective is then to minimize the divergence between the weighted replay buffer's distribution (i.e., the state-action distribution induced by the ensemble) and the expert demonstrator's distribution, making the divergence minimization problem an off-policy learning problem. Similar to boosting and gradient boosting, at every iteration, we aim to find a weak learner, such that when added to the ensemble, the divergence between the updated ensemble's distribution and the expert's distribution decreases. In other words, our approach can be understood as performing gradient boosting in the state-action occupancy space, where black-box RL optimizer is used a weak learning procedure to train weak learners, i.e., policies.

We evaluate `AILBoost` on the DeepMind Control Suite (Tassa et al., 2018) and compare against a range of off-policy AIL algorithms (Behavior cloning, `ValueDICE`, `DAC`) as well as a state-of-the-art IL algorithm, `IQ-Learn`. We show that our algorithm is comparable to or more sample efficient than state-of-the-art IL algorithms in various continuous control tasks, achieving strong imitation performance with as little as one expert demonstration. We also show that our approach scales to vision-based, partially observable domains, where we again outperform `DAC`.

## 2 RELATED WORKS

**Off-policy and Offline IL** There has also been a wide variety of research conducted on off-policy and offline IL, where the goal is to be either more sample efficient or safer by utilizing a replay buffer or not collecting any environmental transitions during training, respectively. The most prominent of said methods, and the closest to our work, is Discriminator-Actor-Critic (`DAC`) (Kostrikov et al., 2019), which essentially replaces the on-policy RL algorithm in the adversarial IL setup with an off-policy one such as DDPG (Lillicrap et al., 2019) or SAC (Haarnoja et al., 2018). However, as mentioned previously, `DAC` doesn't necessarily guarantee a distribution match between the expert and the learned policy, prompting further work to be done. Further work has primarily focused on weighting on-policy and off-policy data differently in both the policy update and the discriminator update. `ValueDICE` (Kostrikov et al., 2020) mitigates this problem by deriving an objective from the original distribution matching problem that only requires off-policy samples to compute. More recently, methods such as `IQ-Learn` (Garg et al., 2021) have been developed to learn soft Q functions over the environment space, which encodes both a reward and a policy for inverse reinforcement learning, and model-based methods such as `V-MAIL` (Rafailov et al., 2021) have shown that using expressive world models (Hafner et al., 2020) leads to strong imitation results in

---

[1]One cannot derive an unbiased estimate of the objective function proposed in `ValueDICE` unless it has infinite expert samples and the transition is deterministic (Kostrikov et al., 2020). See section 3.3 for more detailed discussion.

domains with high-dimensional observations. Other off-policy IL works include SoftDICE (Sun et al., 2021), SparseDICE (Camacho et al., 2021), and AdVIL/AdRIL/DAeQuIL (Swamy et al., 2021).

Orthogonally, on the offline side, where environment interaction is prohibited, works both on the model-based side (Chang et al., 2021) and the model-free side (Kim et al., 2022; Yu et al., 2023) has shown that distribution matching is still possible in these settings. These approaches generally operate either by learning a transition model of the environment, with which to roll out in to do policy optimization (Chang et al., 2021), or optimizing a modified version of the objective introduced in (Kostrikov et al., 2020) by using samples from the suboptimal offline dataset as opposed to on-policy samples for computation.

**Boosting style approach in deep learning & RL** The idea of using boosting for policy learning is not new in the deep learning or reinforcement learning literature. On the deep learning side, AdaGAN (Tolstikhin et al., 2017) apply standard adaptive boosting to GANs (Goodfellow et al., 2014) to address and fix issues such as mode collapse, while concurrent work (Grover & Ermon, 2017) showed benefits of boosting in general Bayesian mixture models. In RL, the conservative policy iteration (CPI) (Kakade & Langford, 2002) can be understood as performing gradient boosting in the policy space (Scherrer & Geist, 2014). The authors in (Hazan et al., 2019) use a gradient boosting style approach to learn maximum entropy policies. In this work, we perform gradient boosting in the space of state-action occupancy measures, which leads to a principled off-policy IL approach.

# 3 PRELIMINARIES

We consider a discounted infinite horizon MDP $\mathcal{M} = \langle \mathcal{S}, P, \mathcal{A}, r, \gamma, \mu_0 \rangle$ where $\mathcal{S}$ is the state of states, $\mathcal{A}$ is the set of actions, $r : \mathcal{S} \times \mathcal{A} \mapsto \mathbb{R}$ is the reward function and $r(s, a)$ is the reward for the given state-action pair, $\gamma \in (0, 1)$ is the discount factor, $\mu_0 \in \Delta(\mathcal{S})$ is the initial state distribution, and $P : \mathcal{S} \times \mathcal{A} \mapsto \Delta(\mathcal{S})$ is the transition function. A policy $\pi : \mathcal{S} \to \Delta(\mathcal{A})$ interacts in said MDP, creating *trajectories* $\tau$ composed of state-action pairs $\{(s_t, a_t)\}_{t=1}^{T}$. We denote $d_t^\pi$ to represent the state-action visitation distribution induced by $\pi$ at timestep $t$ and $d^\pi = (1 - \gamma) \sum_{t=0}^{\infty} \gamma^t d_t^\pi$ as the average state-action visitation distribution induced by policy $\pi$. We define the value function and $Q$-function of our policy as $V^\pi(s) = \mathbb{E}_\pi[\sum_{t=0}^{\infty} \gamma^t r(s_t)|s_0 = s]$ and $Q^\pi(s, a) = r(s, a) + \mathbb{E}_{s' \sim P(\cdot|s,a)}[V^\pi(s')]$. The goal of RL is to find a policy that maximizes the expected cumulative reward.

In imitation learning, instead of having access to the reward function, we assume access to demonstrations $\mathcal{D}^e = \{(s_i, a_i)\}_{i=1}^{N}$ from an expert policy $\pi^e$ that our policy can take advantage of while training. Note that $\pi^e$ might not necessarily be a Markovian policy. It is possible that $\pi^e$ is an ensemble of weighted Markovian policies, i.e., $\pi^e = \{\alpha_i, \pi_i\}_{i=1}^{n}$ with $\alpha_i \geq 0, \sum_i \alpha_i = 1$, which means that for each episode, $\pi^e$ will first randomly sample a policy $\pi_i$ with probability $\alpha_i$ at $t = 0$, and then execute $\pi_i$ for the entire episode (i.e., no switch to other policies during the execution for an episode). It is well known that the space of state action distributions induced by such ensembles is larger than the space of state-action distributions induced by Markovian policies (Hazan et al., 2019). The goal in IL is then to learn a policy that robustly mimics the expert. The simplest imitation learning algorithm to address this issue is behavior cloning (BC): $\text{argmin}_{\pi \in \Pi} \mathbb{E}_{(s,a) \sim \mathcal{D}^e}[\ell(\pi(s), a)]$ where $\ell$ is a classification loss and $\Pi$ is our policy class. Though this objective is simple, it is known to suffer from *covariate shift* at test time (Pomerleau, 1988; Ross et al., 2011). Instead of minimizing action distribution divergence conditioned on expert states, algorithms such as inverse RL (Ziebart et al., 2008) and adversarial IL (Ho & Ermon, 2016; Finn et al., 2016; Ke et al., 2020; Sun et al., 2019) directly minimize some divergence metrics between state-action distributions, which help address the covariate shift issue (Agarwal et al., 2019).

## 3.1 ADVERSARIAL IMITATION LEARNING (AIL)

The goal of AIL is to directly minimize some divergence between some behavior policy state-action visitation $d^\pi$ and an expert policy state-action visitation $d^{\pi^e}$. The choice of divergence results in variously different AIL algorithms.

The most popular AIL algorithm is Generative Adversarial Imitation Learning (GAIL) (Ho & Ermon, 2016) which minimizes the JS-divergence. This algorithm is a on-policy adversarial imitation

learning algorithm that connects Generative Adversarial Networks (GANs) (Goodfellow et al., 2014) and maximum entropy IRL (Ziebart et al., 2008). `GAIL` trains a binary classifier called the discriminator $D(s, a)$ to distinguish between samples from the expert distribution and the policy generated distribution. Using the discriminator to define a reward function, `GAIL` then executes an on-policy RL algorithm such as Trust Region Policy Optimization (TRPO) (Schulman et al., 2017a) or Proximal Policy Optimization (PPO) (Schulman et al., 2017b) to maximize the reward. That gives us the following adversarial objective:

$$\min_{\pi} \max_{D} \mathbb{E}_{s,a \sim \pi} \left[ \log D(s, a) \right] + \mathbb{E}_{s,a \sim \pi^e} \left[ \log(1 - D(s, a)) \right] - \lambda H(\pi) \tag{1}$$

where $H(\pi)$ is an entropy regularization term. The first term in eq. (1) can be viewed as a pseudo reward that can be optimized with respect to the the policy $\pi$ on-policy samples. Note that GAIL typically optimizes both policies and discriminators using on-policy samples, making it quite sample inefficient. Using different divergences, there are various reward functions that can be optimized with this framework (Orsini et al., 2021). In this work, while our proposed approach in general is capable of optimizing many common divergences, we mainly focus on reverse KL divergence in our experiments. Reverse KL divergence has been studied in prior works including Fu et al. (2018); Ke et al. (2020). But different from prior works, we propose an off-policy method for optimizing reverse KL by leveraging the framework of boosting.

## 3.2 DISCRIMINATOR ACTOR CRITIC (`DAC`)

One reason `GAIL` need a lot of interactions with the environment to learn properly is because of the dependency on using on-policy approaches to optimize discriminators and policies. In particular, `GAIL` does not reuse any old samples. Discriminator Actor Critic (`DAC`) (Kostrikov et al., 2019) extends `GAIL` algorithms to take advantage of off-policy learning to optimize the discriminators and policies.

`DAC` introduces a replay buffer $\mathcal{R}$ to represent the history of transitions observed throughout training in the context of IRL. This replay buffer allows `DAC` to perform off-policy training of the policy and the discriminator (similar to (Sasaki et al., 2019)). Formally, `DAC` optimizes its discriminator with the objective:

$$\max_{D} \mathbb{E}_{s,a \sim \mathcal{R}} \left[ \log D(s, a) \right] + \mathbb{E}_{s,a \sim \pi^e} \left[ \log(1 - D(s, a)) \right]. \tag{2}$$

where this objective minimize the divergence between the expert distribution and the replay buffer $\mathcal{R}$ distribution. Intuitively, this divergence does not strictly capture the divergence of our policy distribution and the expert distribution, but a mixture of evenly weighted policies learned up until the current policy. To rigorously recover a divergence between our policy distribution and the expert distribution we need to apply importance weights: $\min_{\pi} \max_{D} \mathbb{E}_{s,a \sim \mathcal{R}} \left[ \frac{p_{\pi}(s,a)}{p_{\mathcal{R}}(s,a)} \log D(s, a) \right] + \mathbb{E}_{s,a \sim \pi^e} \left[ \log(1 - D(s, a)) \right] - \lambda H(\pi)$. While this objective recovers the *on-policy* objective of `GAIL` (Equation (1)), the authors note that estimating the density ratio is difficult and has high variance in practice. Furthermore, they note that the not using importance weights (Equation (2)) works well in practice, *but does not guarantee successful imitation*, especially when the distribution induced by the replay buffer, $\mathcal{R}$, is far from our current policy's state-action distribution. This is a fundamental problem of DAC.

## 3.3 VALUEDICE

`ValueDICE` (Kostrikov et al., 2020) was proposed to address the density estimation issue of off-policy AIL algorithms formalized in `DAC` (see section 3.2). `ValueDICE` aims to minimize the reverse KL divergence written in its Donsker-Varadhan (Donsker & Varadhan, 1983) dual form:

$$-\text{KL}(d^{\pi} || d^{\pi_e}) = \min_{x: \mathcal{S} \times \mathcal{A} \mapsto \mathbb{R}} \log \mathbb{E}_{(s,a) \sim d^{\pi_e}} \left[ e^{x(s,a)} \right] - \mathbb{E}_{(s,a) \sim d^{\pi}} \left[ x(s, a) \right] \tag{3}$$

Motivated from `DualDICE` (Nachum et al., 2019), `ValueDICE` performs a change of variable using the Bellman operator $\mathcal{B}^{\pi\,2}$ with respect to the policy $\pi$; $x(s, a) = \nu(s, a) - \mathcal{B}^{\pi}(s, a)$; resulting

---

[2] A bellman operator $\mathcal{B}^{\pi}$ is defined as follows: given any function $f(s, a)$, we have $\mathcal{B}^{\pi} f(s, a) := r(s, a) + \mathbb{E}_{s' \sim P(s,a)} f(s', \pi(s')), \forall s, a$.

the following objective:

$$\max_{\pi} \min_{\nu: \mathcal{S} \times \mathcal{A} \to \mathbb{R}} \log \mathbb{E}_{s,a \sim \pi^e} \left[ \exp \left( \nu(s,a) - \mathcal{B}^{\pi} \nu(s,a) \right) \right] - (1 - \gamma) \mathbb{E}_{\substack{s_0 \sim \mu_0, \\ a_0 \sim \pi}} \left[ \nu(s_0, a_0) \right]. \quad (4)$$

Now the objective function does not contain on-policy distribution $d^{\pi}$ (in fact only the initial state distribution $\mu_0$ and the expert distribution). Despite being able to only using $d^{\pi^e}$ and $\mu_0$, the authors have identified two aspects of the objective that will yield biased estimates. First, the first expectation has a logarithm outside of it which would make mini-batch estimates of this expectation biased. Moreover, inside the first expectation term, we have $\nu(s,a) - \mathcal{B}^{\pi} \nu(s,a)$ with $\mathcal{B}^{\pi}$ being the Bellman operator. This limits `ValueDICE`'s objective to only be unbiased for environments with deterministic transitions. This is related to the famous double sampling issue in TD learning. Although many popular RL benchmarks have deterministic transitions (Bellemare et al., 2013; Tassa et al., 2018; Todorov et al., 2012), this was a limitation not present in the `GAIL`.

In this work, we take a different perspective than `ValueDICE` to derive an off-policy AIL algorithm. Different from `ValueDICE`, our approach is both off-policy and is amenable to mini-batch updates even with stochastic environment transition dynamics.

## 4 ALGORITHM

Our algorithm, Adversarial Imitation Learning via Boosting (`AILBoost`) – motivated by classic *gradient boosting* algorithms (Friedman, 2001; Mason et al., 1999) – attempts to mitigate a fundamental issue related to off-policy imitation learning formalized in `DAC` (see section 3.2). The key idea is to treat learned policies as weak learners, form an ensemble of them (with a proper weighting scheme derived from a gradient boosting perspective), and update the ensemble via gradient boosting.

**Weighted policy ensemble.** Our algorithm will learn a weighted ensemble of policies, denoted as $\boldsymbol{\pi} := \{\alpha_i, \pi_i\}_{i=1}^n$ with $\alpha_i \geq 0, \sum_i \alpha_i = 1$ and $\pi_i$ being some Markovian policy. The way the mixture works is that when executing $\boldsymbol{\pi}$, at the beginning of an episode, a Markovian policy $\pi_i$ is sampled with probability $\alpha_i$, and then $\pi_i$ is executed for the entire episode (i.e., no policy switch in an episode). Note that $\boldsymbol{\pi}$ itself is not a Markovian policy anymore due to the sampling process at the beginning of the episode, and in fact, such mixture policy's induced state-action distribution can be richer than that from Markovian policies (Hazan et al., 2019). This is consistent with the idea of *boosting*: by combining weak learners, i.e., Markovian policies, we form a more powerful policy. Given the above definition of $\boldsymbol{\pi}$, we immediately have $d^{\boldsymbol{\pi}} := \sum_i \alpha_i d^{\pi_i}$, i.e., the weighted mixture of the state-action distributions induced by Markovian policies $\pi_i$.

Notation wise, given a dataset $\mathcal{D}$, we denote $\widehat{\mathbb{E}}_{\mathcal{D}}[f(x)]$ as the empirical function average across the dataset, i.e., $\widehat{\mathbb{E}}_{\mathcal{D}}[f(x)] = \sum_{x \in \mathcal{D}} f(x) / |\mathcal{D}|$.

### 4.1 `AILBoost`: ADVERSARIAL IMITATION LEARNING VIA BOOSTING

We would like to minimize the reverse KL divergence between our policy state-action distribution $d^{\boldsymbol{\pi}}$ and the expert distribution $d^{\pi^e}$ – denoted by $\ell(d^{\boldsymbol{\pi}}, d^{\pi^e}) = \mathrm{KL}(d^{\boldsymbol{\pi}} || d^{\pi^e}) := \sum_{s,a} d^{\boldsymbol{\pi}}(s,a) \ln(d^{\boldsymbol{\pi}}(s,a)/d^{\pi^e}(s,a))$. The reasons that we focus on reverse KL is that (1) it has been argued that the mode seeking property of reverse KL is more suitable for imitation learning (Ke et al., 2020), (2) reverse KL is on-policy in nature, i.e., it focuses on minimizing the divergence of our policy's action distribution and the expert's at the states from our policy, which help address the covariate shift issue, and (3) the baselines we consider in experiments, `DAC` and `ValueDICE`, all minimize the reverse KL divergence such as `AIRL` in practice [3]. At a high level, our approach directly optimizes $\ell(d^{\boldsymbol{\pi}}, d^{\pi^e})$ via gradient boosting (Mason et al., 1999) in the state-action occupancy space. Our ensemble $\boldsymbol{\pi}$ induces the following mixture state-action occupancy measure:

$$d^{\boldsymbol{\pi}} := \sum_{i=1}^{t} \alpha_i d^{\pi_i}, \alpha_i \geq 0.$$

To compute a new weak learner $\pi_{t+1}$, we will first compute the functional gradient of loss $\ell$ with respect to $d^{\boldsymbol{\pi}}$, i.e., $\nabla \ell(d, d^{\pi^e})|_{d=d^{\boldsymbol{\pi}}}$. The new weak learner $\pi_{t+1}$ is learned via the following

---

[3]See the official repository

---

**Algorithm 1** AILBOOST (**A**dversarial **I**mitation **L**earning via **Boost**ing)

**Require:** number of iterations $T$, expert data $\mathcal{D}^e$, weighting parameter $\alpha$
1: Initialize $\pi_1$ weight $\alpha_1 = 1$, replay buffer $\mathcal{B} = \emptyset$
2: **for** $t = 1, \dots, T$ **do**
3:     Construct the $t$-th dataset $\mathcal{D}_t = \{(s_j, a_j)\}_{j=1}^N$ where $s_j, a_j \sim d^{\pi_t} \; \forall j$.
4:     Compute discriminator $\hat{g}$ using the weighted replay buffer:

$$\hat{g} = \arg\max_g \left[ \widehat{\mathbb{E}}_{s,a \in \mathcal{D}^{\pi^e}} \left[ -\exp(g(s,a)) \right] + \textcolor{red}{\sum_{i=1}^t \alpha_i \widehat{\mathbb{E}}_{s,a \in \mathcal{D}_i} \left[ g(s,a) \right]} \right] \tag{5}$$

5:     Set $\mathcal{B} \leftarrow \mathcal{B} \cup \mathcal{D}_t$
6:     Compute weak learner $\pi_{t+1}$ via an off-policy RL approach (e.g., SAC) on reward $-\hat{g}(s,a)$
       with replay buffer $\mathcal{B}$
7:     Set $\alpha_i \leftarrow \alpha_i(1-\alpha)$ for $i \le t$, and $\alpha_{t+1} = \alpha$
8: **end for**
9: **Return** Ensemble $\boldsymbol{\pi} = \{(\alpha_i, \pi_i)\}_{i=1}^T$

---

optimization procedure: $\pi_{t+1} = \arg\max_{\tilde{\pi} \in \Pi} \langle d^{\tilde{\pi}}, -\nabla \ell(d, d^{\pi^e})|_{d=d^{\boldsymbol{\pi}}} \rangle$. Namely, we aim to search for a new policy $\pi_{t+1}$ such that its state-action occupancy measure $d^{\pi_{t+1}}$ is aligned with the negative gradient $-\nabla \ell$ as much as possible. Note that the above optimization problem can be understood as an RL procedure where the reward function is defined as $-\nabla \ell(d, d^{\pi^e})|_{d=d^{\boldsymbol{\pi}}} \in \mathbb{R}^{SA}$. Once we compute the weak learner $\pi_{t+1}$, we mix it into the policy ensemble with a fixed learning rate $\alpha \in (0,1)$ – denoted as $d^{\boldsymbol{\pi}'} = (1-\alpha)d^{\boldsymbol{\pi}} + \alpha d^{\pi_{t+1}}$. Note that the above mixing step can be interpreted as gradient boosting in the state-action occupancy space directly: we re-write the update procedure as $d^{\boldsymbol{\pi}'} = d^{\boldsymbol{\pi}} + \alpha(d^{\pi_{t+1}} - d^{\boldsymbol{\pi}})$, where the ascent direction $d^{\pi_{t+1}} - d^{\boldsymbol{\pi}}$ is approximating the (negative) functional gradient $-\nabla \ell$, since $\arg\max_\pi \langle d^\pi - d^{\boldsymbol{\pi}}, -\nabla \ell \rangle = \pi_{t+1}$ by the definition of $\pi_{t+1}$. It has been shown that such procedure is *guaranteed to minimize the objective function* (i.e., reverse KL in this case) as long as the objective is smooth (our loss $\ell$ will be smooth as long as $d^{\boldsymbol{\pi}}$ is non-zero everywhere) (e.g., see (Hazan et al., 2019) for the claim).[4]

Algorithmically, we first express the reverse KL divergence in its variational form (Nowozin et al., 2016; Ke et al., 2020):

$$\text{KL}(d^{\boldsymbol{\pi}} || d^{\pi^e}) := \max_g \left[ \mathbb{E}_{s,a \sim d^{\pi^e}} \left[ -\exp(g(s,a)) \right] + \mathbb{E}_{s,a \sim d^{\boldsymbol{\pi}}} g(s,a) \right]$$

where $g : \mathcal{S} \times \mathcal{A} \mapsto \mathbb{R}$ is a discriminator. The benefit of using this variational form is that computing the functional (sub-)gradient of the reverse KL with respect to $d^{\boldsymbol{\pi}}$ is easy, which is $\hat{g} = \arg\max_g \left[ \mathbb{E}_{s,a \sim d^{\pi^e}} \left[ -\exp(g(s,a)) \right] + \mathbb{E}_{s,a \sim d^{\boldsymbol{\pi}}} g(s,a) \right]$, i.e., we have $\hat{g}$ being a functional sub-gradient of the loss $\text{KL}(d^{\boldsymbol{\pi}} || d^{\pi^e})$ with respect to $d^{\boldsymbol{\pi}}$. The maximum discriminator $\hat{g}$ will serve as a reward function for learning the next weak learner $\pi_{t+1}$, that is

$$\pi_{t+1} = \arg\max_\pi \mathbb{E}_{s,a \sim d^\pi} \left[ -\hat{g}(s,a) \right] = \arg\max_\pi \langle d^\pi, -\hat{g}(s,a) \rangle. \tag{6}$$

To compute $\hat{g}$ in practice, we need unbiased estimates of the expectations via sample averaging which can be done easily in our case. The expectation $\mathbb{E}_{s,a \sim d^{\pi^e}}$ can be easily approximated by the expert dataset $\mathcal{D}^e$. To approximate $\mathbb{E}_{s,a \sim d^{\boldsymbol{\pi}}}$ where $d^{\boldsymbol{\pi}}$ is a mixture distribution, we maintain a replay buffer $\mathcal{D}_i$ for each weak learner $\pi_i$ which contains samples $s, a \sim d^{\pi_i}$, and then weight $\mathcal{D}_i$ via the weight $\alpha_i$ associated with $\pi_i$. In summary, we optimize $g$ as shown in Eq. 5 in Alg 4.1 (the highlighted red part denotes the empirical expectation induced by weighted replay buffer). The optimization problem in Eq 5 can be solved via stochastic gradient ascent on $g$.[5] With $\hat{g}$, we can optimize for $\pi_{t+1}$ using any off-shelf RL algorithm, making the entire algorithm off-policy. In our experiments, we use SAC as the RL oracle for $\arg\max_\pi \mathbb{E}_{s,a \sim d^\pi}[-\hat{g}(s,a)]$. Once $\pi_{t+1}$ is computed, we mix $\pi_{t+1}$ into the mixture,

---

[4]Note that similar to AdaBoost, each weaker is not directly optimizing the original objective, but the weighted combination of the weaker learners optimizes the original objective function – the reverse KL in our case.

[5]Note that unlike ValueDICE, here we can easily use a finite number of samples to obtain an unbiased estimate of the loss by replacing expectations by their corresponding sample averages.

and adjust the weights of older policies accordingly, i.e., $\alpha_{t+1} = \alpha$, and $\alpha_i \leftarrow \alpha_i(1 - \alpha), \forall i \leq t$. Note that this weighting scheme ensures that older policies get less weighted in the ensemble.

**Remark 1.** *The use of SAC as the weak learning algorithm and the new way of computing discriminator from Eq. 5 make the whole training process completely off-policy. Particularly, unlike most adversarial IL approaches, which compute discriminators by comparing on-policy samples from the latest policy and the expert samples, we train the discriminator using all the data collected so far (with proper weighting derived based on the boosting framework). The connection to boosting and the proper weighting provides a principled way of leveraging off-policy samples for updating discriminators. As we will show, compared to* DAC *which also uses off-policy samples for training policies and discriminators, our principled approach leads to better performance.*

Alg 4.1 `AILBoost`, summarizes the above procedure. In Line 10, we use SAC as the RL oracle for computing the weak learner. In practice, we do not run SAC from scratch every time in Line 10. Instead, SAC maintains its own replay buffer which contains all interactions it has with the environment so far. When computing $\pi_{t+1}$, we first update the reward in the replay buffer using the latest learned reward function $-\hat{g}$, and we always warm start from $\pi_t$. We include the detailed algorithmic description in Appendix A.

**Memory cost.** Note that at the end, our algorithm returns a weighted ensemble of Markovian policies. Comparing to prior works such as DAC, the maintenance of weak learners may increase additional memory cost. However, the benefit of the weighted ensemble is that it induces richer state-action distributions than that of Markovian policies. In practice, if memory cost really becomes a burden (not in our experiments even with image-based control policies), we may just keep the latest few policies (note that very old policy has exponentially small weight anyway).

## 5 EXPERIMENTS

In this section we aim to empirically investigate the following questions: (1) How does `AILBoost` perform relative to other off-policy and state-of-the-art IL methods? (2) Does `AILBoost` enjoy the sample complexity and scalability benefits of modern off-policy IL methods? (3) How robust is `AILBoost` across various different adversarial training schedules?

We evaluate `AILBoost` on 5 environments on the DeepMind Control Suite benchmark(Tassa et al., 2018): `Walker Walk`, `Cheetah Run`, `Ball in Cup Catch`, `Quadruped Walk`, and `Humanoid Stand`. For each game, we train an expert RL agent using the environment's reward and collect 10 demonstrations which we use as the expert dataset throughout our experiments. We compare `AILBoost` against the following baselines: `DAC`, an empirically succesful off-policy IL algorithm; `IQ-Learn`, a state-of-the-art IL algorithm; `ValueDICE`, another off-policy IL method; and BC on the expert data used across all algorithms. We emphasize our comparison to `IQ-Learn`, as it has been shown to outperform many other imitation learning baselines (e.g., SQIL (Reddy et al., 2019)) across a variety of control tasks (Garg et al., 2021).

| Task | Difficulty |
|---|---|
| Ball in Cup Catch | Easy |
| Walker Walk | Easy |
| Cheetah Run | Medium |
| Quadruped Walk | Medium |
| Humanoid Stand | Hard |

Table 1: Spread of environments evaluated from the DeepMind Control Suite with hardness designations from (Yarats et al., 2022).

The base RL algorithm we used for training the expert, as well as for `AILBoost` and `DAC`, was SAC for controller state-based experiments and DrQ-v2 (Yarats et al., 2022) for image-based experiments. For `IQ-Learn` and `ValueDICE`, we used their respective codebases and hyperparameters provided by the authors and both methods use SAC as their base RL algorithm. Please refer to Appendix B for experimental details, training hyperparameters, and expert dataset specifications.

### 5.1 CONTROLLER STATE-BASED EXPERIMENTS

Figure 1 shows our aggregate results across the five DeepMind Control Suite (DMC) tasks that we tested on. We chose these five tasks by difficulty as shown in Table 1. For evaluation, we follow the recommendations of (Agarwal et al., 2021) and report the aggregate inter-quartile mean, mean, and optimiality gap of `AILBoost` and all the baselines on the DMC suite with 95% confidence intervals.

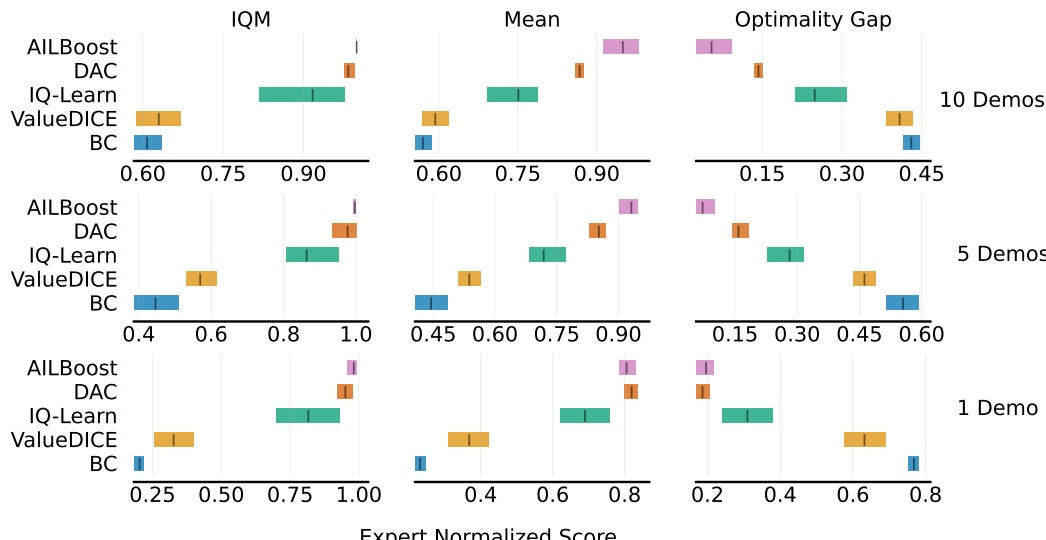

Figure 1: **Aggregate metrics on DMC environments** with 95% confidence intervals (CIs) based on 5 environments spanning *easy*, *medium*, and *hard* tasks. Higher inter-quartile mean (IQM) and mean scores (right) and lower optimality gap (left) is better. The CIs were calculated with percentile bootstrap with stratified sampling over three random seeds and all metrics are reported on the expert normalized scores. **AILBoost outperforms `DAC`, `ValueDICE`, `IQ-Learn`, and `BC` across all metrics, amount of expert demonstrations, and tasks**.

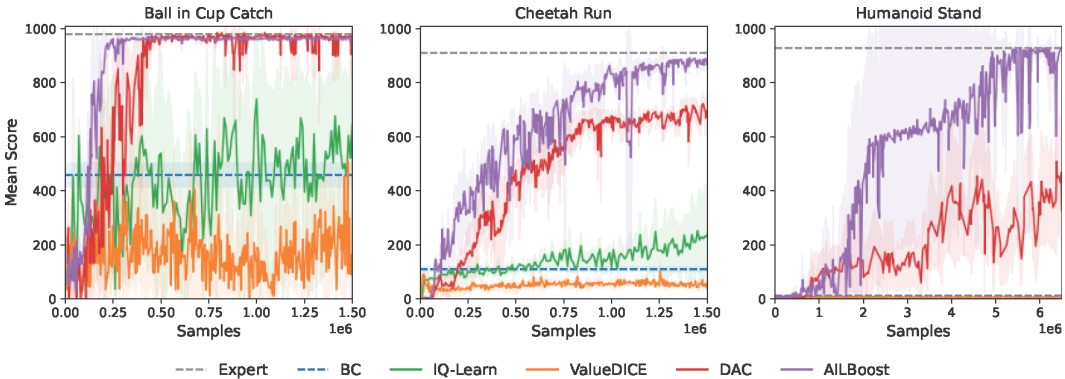

Figure 2: **Learning curves with 1 expert trajectory** across 3 random seeds. **Note `AILBoost` successfully imitates expert on all environments where other baselines fail and achieves better sample complexity than `DAC`**. Note that when the environment difficulty level increases, our method shows a larger performance gap compared to baselines (e.g., humanoid stand).

We find that `AILBoost` not only outperforms all baselines but also consistently matches the expert with only 1 expert trajectory.

When we inspect the 1 trajectory case closer, Figure 2 shows the learning curves on three representative (1 easy, 1 medium, 1 hard task) environments where we see `AILBoost` maintain high sample efficiency and strong imitation while state-of-the-art baselines like `IQ-Learn` completely fail on `Humanoid Stand`. Finally, we note that `AILBoost` greatly outperforms `ValueDICE` which aimed to make AIL off-policy from a different perspective. We refer readers to Figure 6 in the appendix for the learning curves on all five environments with different numbers of expert demonstrations.

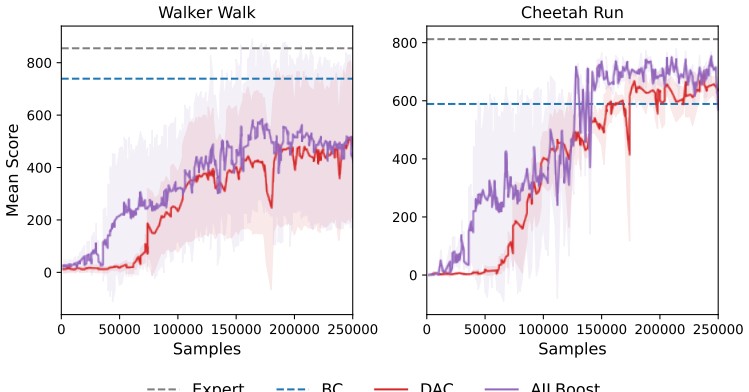

Figure 3: **Image based:** performance on image-based DMC environments, `Walker Walk` and `Cheetah Run`, comparing `AILBoost`, `DAC`, and `BC` on three random seeds.

## 5.2 IMAGE-BASED EXPERIMENTS

Figure 3 demonstrates the scalability of `AILBoost` on a subset of environments with 10 expert trajectories. For these experiments, we use DrQ-v2 (Yarats et al., 2022) as the underlying off-policy RL algorithm for both `DAC` and `AILBoost`. On `Walker Walk` and `Cheetah Run`, we see comparable to better performance than `DAC` demonstrating that our boosting strategy successfully maintains the empirical, scaling properties of `DAC`. Furthermore, our use of different off-policy RL algorithms show the versatility of `AILBoost` for IL.

## 5.3 SENSITIVITY TO GRADIENT-BASED OPTIMIZATION FOR WEAK LEARNERS AND DISCRIMINATORS

Our algorithm relies on solving optimization problems in Eq. 6 and Eq. 5 for weak learners and discriminators, where weak learner is optimized by SAC and discriminators are optimized by SGD. While it is hard to guarantee in general that we can exactly solve the optimization problem due to our policies and discriminators are both being non-convex neural networks, we in general found that approximately solving Eq. 6 and Eq. 5 via gradient based update is enough to ensure good performance. In this section, we test `AILBoost` across a variety of optimization schedules. Overall, we find that `AILBoost` to be robust to optimization schedules — approximately optimizing Eq. 6 and Eq. 5 with sufficient amount of gradient updates ensures successful imitation; however, there exists a sample complexity cost when over-optimizing either the discriminator or the policy.

Figure 4 shows our investigation of how sensitive `AILBoost` is to different optimization schedules for both the policy and discriminator on two representative DMC environments. In particular, we test with 5 expert demonstrations, where we vary the number of discriminator and policy updates. We test the following update schemes:

- 1000 policy updates per 100 discriminator updates
- 1000 policy updates per 10 discriminator updates
- 1000 policy updates per 1 discriminator update
- 100 policy updates per 100 discriminator updates

These ranges, test various optimization schemes around the schedule that we chose for the main results. We find that the more policy updates we do per discriminator update, the algorithm becomes significantly less sample efficient despite asymptotically reaching expert performance. We also found that an insufficient amount of updates on the discriminator general hurts the performance. This is also expected since insufficient update on the discriminators may result a $\hat{g}$ which does not optimize Eq. 5 well enough.

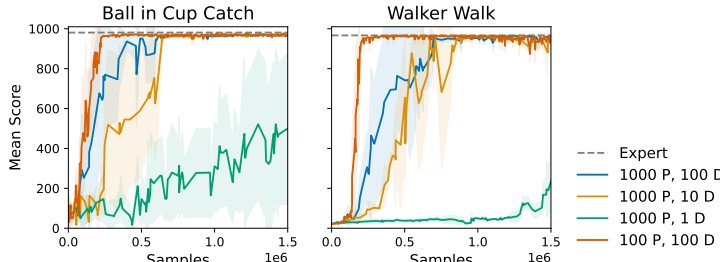

Figure 4: **Policy and Discriminator Update Schedules:** Learning curves for `AILBoost` on two representative DMC environments, `Walker Walk` and `Ball in Cup Catch`, when optimizing with varying policy and discriminator update schemes across 3 seeds.

## 6 CONCLUSION

In this work, we present a fully off-policy adversarial imitation learning algorithm, `AILBoost`. Different from previous attempts at making AIL off-policy, via the gradient boosting framework, `AILBoost` provides a principled way of re-using old data for learning discriminators and policies. We show that our algorithm achieves state-of-the-art performance on state-based results on the DeepMind Control Suite while being able to scale to high-dimensional, pixel observations. We are excited to extend this framework to discrete control as well as investigate imitation learning from observations alone under this boosting framework.

## ACKNOWLEDGEMENTS

We would like to acknowledge the support of NSF under grant IIS-2154711, NSF CAREER 2339395, and Cornell Infosys Collaboration. Jonathan Chang is supported by LinkedIn under the LinkedIn-Cornell Grant. Kiante Brantley is supported by NSF under grant No. 2127309 to the Computing Research Association for the CIFellows Project.

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

## A  DETAILED ALGORITHM PSEUDOCODE

Appendix A presents a more detailed pseudocode of `AILBoost`. The main detail here is the 2-step process of learning our discriminator using a weighted replay buffer of weak learner samples and then learning a weak learner for a certain number of RL steps.

---

**Algorithm 2** AILBOOST (**A**dversarial **I**mitation **L**earning via **Boost**ing)

---

**Require:** number of iterations $T$, expert data $\mathcal{D}^e$, weighting parameter $\alpha$
1: Initialize $\pi_1$ weight $\alpha_1 = 1$, replay buffer $\mathcal{B} = \emptyset$
2: **for** $t = 1, \ldots, T$ **do**
3:    Construct the $t$-th dataset $\mathcal{D}_t = \{(s_j, a_j)\}_{j=1}^N$ where $s_j, a_j \sim d^{\pi_t} \ \forall j$.
4:    Set $\mathcal{B} \leftarrow \mathcal{B} \cup \mathcal{D}_t$
5:    **for** # of Discriminator Updates **do**
6:       Sample batch from $\mathcal{B}$ with respective sample weights $\alpha_{i<t}$
7:       Update discriminator $\hat{g}$ via Eq. 5
8:    **end for**
9:    **for** # of Weak Learner Updates **do**
10:       Compute weak learner $\pi_{t+1}$ via an off-policy RL approach (e.g., SAC) on reward $-\hat{g}(s, a)$
         with replay buffer $\mathcal{B}$ with uniform weights on all samples
11:    **end for**
12:    Set $\alpha_i \leftarrow \alpha_i(1 - \alpha)$ for $i \leq t$, and $\alpha_{t+1} = \alpha$
13: **end for**
14: **Return** Ensemble $\boldsymbol{\pi} = \{(\alpha_i, \pi_i)\}_{i=1}^T$

---

After learning our ensemble, we evaluate it by randomly sampling a policy, $\pi_i$, from our ensemble with probability $\alpha_i$. With this weighted sampling, we then collect a trajectory. Appendix A details this process.

---

**Algorithm 3** AILBOOST EVALUATION

---

**Require:** Ensemble $\boldsymbol{\pi} = \{(\alpha_i, \pi_i)\}_{i=1}^T$
1: **for** # of Evaluation Trajectories **do**
2:    Sample $\pi_i \sim \boldsymbol{\pi}$ with probability $\alpha_i$
3:    Collect trajectory using $\pi_i$
4: **end for**

---

## B  IMPLEMENTATION AND EXPERIMENT DETAILS

Here we detail all environment specifications and hyperparameters used in the main text.

### B.1  ENVIRONMENT DETAILS

Following the standards used by DrQ-v2 (Yarats et al., 2022), all environments have a maximum horizon length of 500 timesteps. This is achieved by setting each environment's action repeat to be 2 frames. For image based tasks, each state is 3 stacked frames that are each $84 \times 84$ dimensional RGB images (thus $9 \times 84 \times 84$).

| Task | Action Space Dimension | Task Traits | Reward Type |
|------|:----:|:----:|:----:|
| Ball in Cup Catch | 2 | swing, catch | sparse |
| Walker Walk | 6 | walk | dense |
| Cheetah Run | 6 | run | dense |
| Quadruped Walk | 12 | walk | dense |
| Humanoid Stand | 21 | stand | dense |

Table 2: Task descriptions, action space dimension, and reward type for each tested environment.

### B.2  DATASET DETAILS

Using the publicly released implementation for SAC and DrQ-v2, we trained high quality expert policies for state-based and image-based environments respectively. We refer the readers to (Yarats et al., 2022) and (Haarnoja et al., 2018; Yarats & Kostrikov, 2020) for exact hyperparameters.

| Task | Expert Performance | Random Performance |
|------|:----:|:----:|
| Ball in Cup Catch | 980.8 | 16.4 |
| Walker Walk | 966.6 | 19.9 |
| Cheetah Run | 910.5 | 0.2 |
| Quadruped Walk | 959.2 | 17.9 |
| Humanoid Stand | 927.8 | 3.9 |
| Walker Walk (Vision) | 823.1 | 9.6 |
| Cheetah Run (Vision) | 806.3 | 0.3 |

Table 3: Average expert and random performance calculated by averaging 50 trajectories collected from the expert and random policies respectively. Vision experts are denoted (vision)

### B.3  HYPERPARAMETERS

For ValueDICE and IQ-Learn, we used the base hyperparameters they reported for the MuJoCo benchmark suite. In order to ensure good performance, we tried different configurations for every environments (i.e. the configuration for Cheetah Run for Walker Walk) since despite using the same physics engine and models, there are minor differences for DeepMind Control Suite. For DAC and AILBoost, we used our own implementations. Table 4 details the hyperparameters used. Note that all hyperparameters are shared between DAC and AILBoost except for the update frequency of the disciminrator vs the policy. Note that this is one of the core differences between DAC and AILBoost.

For AILBoost we predominanty tested 4 hyperparameters: # of discriminator updates, steps to learn weak learners, weighting parameter $\alpha$, and the TD n-step. For the # of discriminator updates we tested 10, 100, 500, 1000, and 5000. For the the steps to learn weak learners, we tested 1000, 5000, 10000, 20000, and 100000. For $\alpha$, we swept 0.95, 0.7, 0.4, 0.2, and 0.05. Finally, we tested either TD n-step 1 or 3.

| Setting | Values |
|---|---|
| Policy Architecture (state) | 3 layer MLP with 1024 hidden units each |
| `DAC` (state) | total number of steps: 10e6
replay buffer size: 1e6
learning rate: 1e-4
action repeat: 2
batch size: 256
TD n-step: 1
discount factor: 0.99
gradient penalty coeff: 10.0
policy update frequency: 2 |
| `AILBoost` (state) | Samples per Weak Learner (N): 1000
# of Weak Learners (T): 100
Steps to learn Weak Learner: 1000
# of Discriminator updates: 100
Weighting Parameter ($\alpha$): 0.05 |
| Policy Architecture (vision) | Model Architecture from (Yarats et al., 2022) |
| `DAC` (vision) | total number of steps: 20e6
replay buffer size: 1e6
learning rate: 1e-4
action repeat: 2
batch size: 512
TD n-step: 3
discount factor: 0.99
gradient penalty coeff: 10.0
policy update frequency: 2 |
| `AILBoost` (vision) | Samples per Weak Learner (N): 10000
# of Weak Learners (T): 100
Steps to learn Weak Learner: 20000
# of Discriminator updates: 500
Weighting Parameter ($\alpha$): 0.05 |

Table 4: Hyperparameters used for `DAC` and `AILBoost`. All of `DAC`'s hyperparameters are shared by `AILBoost` except for the parameters colored in blue. In particular, the update frequency of the disciminrator vs the policy is one of the core differences between `DAC` and `AILBoost`.

## C    ADDITIONAL RESULTS

### C.1    AGGREGATE PERFORMANCE COMPARISONS

Following the recommendations of (Agarwal et al., 2021), we do an additional diagnostic of measuring the probability of improvement between two algorithms. This metric measures how likely it is for $X$ to outperform $Y$ on a randomly selected task from the benchmark suite. Specifically, $P(X > Y) = \frac{1}{m} \sum_m P(X_m > Y_m)$ where $P(X_m > Y_m)$ is the probability of $X$ outperforming $Y$ on task $m$. Note that this measurement does not account for the *size of improvement*. Figure 5 shows the comparison. AILBoost shows significant improvement over all other algorithms other than DAC. In conjuction with Figure 1, we see that although the chance of AILBoost doing better than DAC is $\approx 50\%$, the size of improvement AILBoost has over DAC denoted by the IQM and Mean are significantly larger.

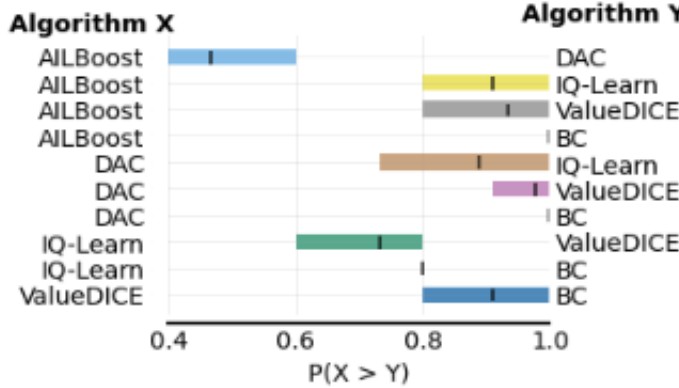

Figure 5: **Probability of improvement** between all tested baselines and AILBoost.

## C.2 LEARNING CURVES

Here we present the complete suite of learning curves for all 5 environments.

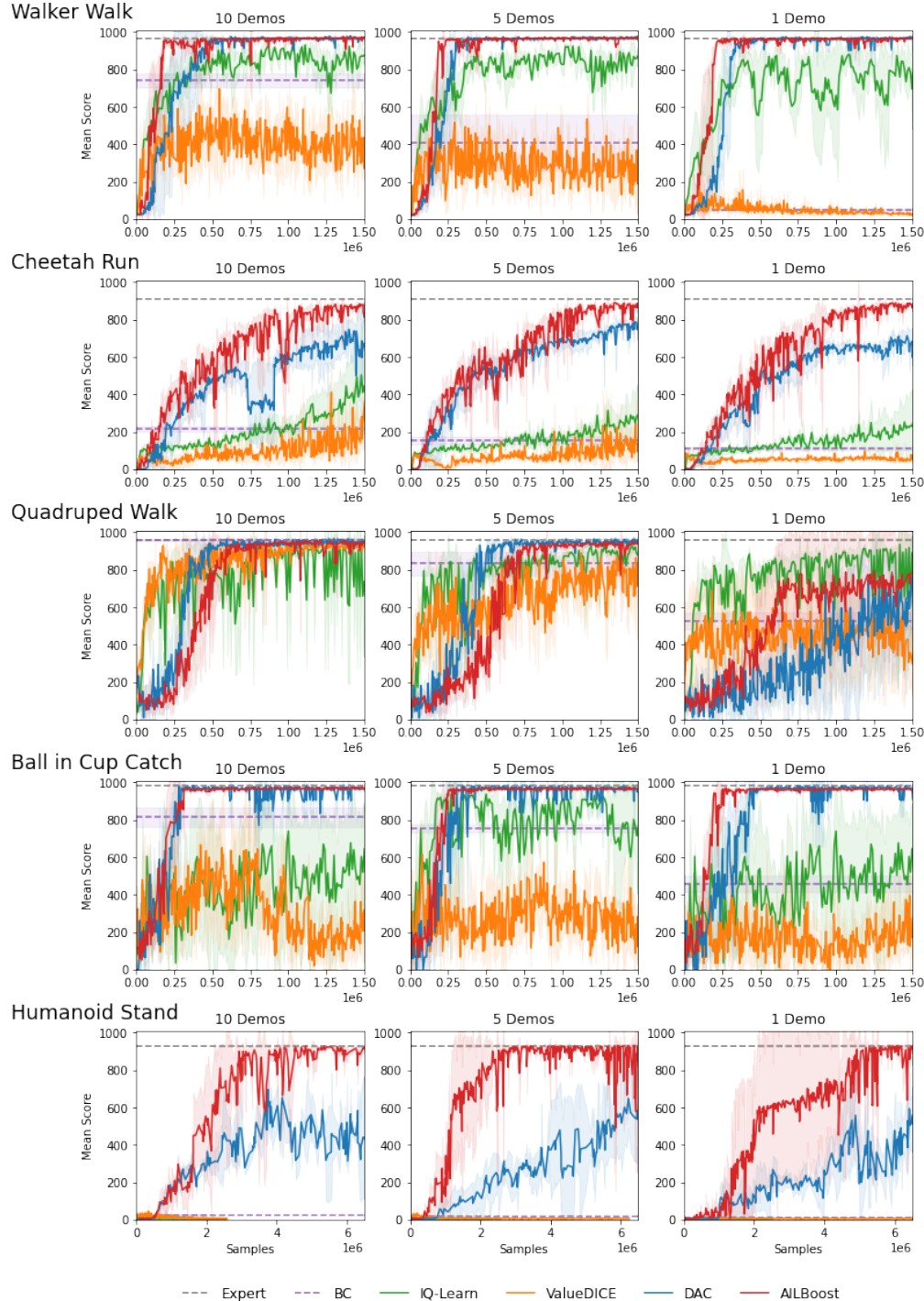

Figure 6: **Learning curves** for `AILBoost` and all baselines on the DMC environments with 10, 5, and 1 expert trajectories across 3 seeds.

## C.3 LEARNING CURVES ACROSS DIFFERENT OPTIMIZATION SCHEDULES

Here we present the full suite of learning curves where we vary how often the policy and the discriminator update relative to each other. We keep every other hyperparameter constant in this ablation.

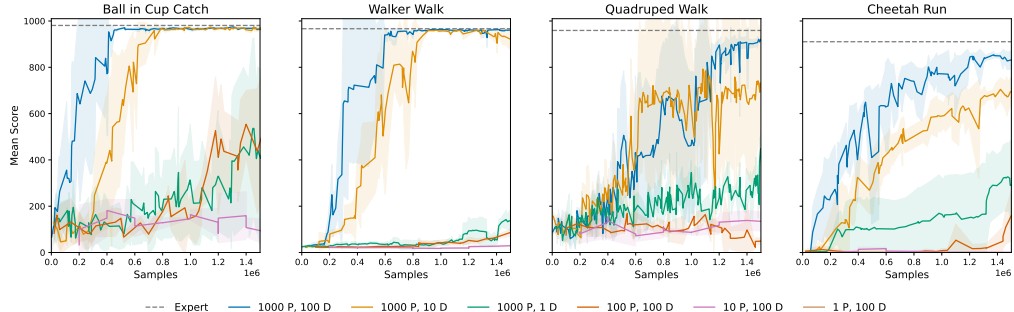

Figure 7: Learning curves for `AILBoost` on 4 out of the 5 DMC environments with 5 expert trajectories across 3 seeds, where we vary the number of policy updates and discriminator updates the agent takes over time.

