# OpenReview forum: "Adversarial Imitation Learning via Boosting"
_ICLR.cc/2024/Conference — ICLR 2024 poster_

### Official Review · Reviewer_xvjX · 2023-10-27

**Soundness:** 3 good
**Presentation:** 3 good
**Contribution:** 3 good
**Rating:** 5
**Confidence:** 3

**Summary:**

This paper proposes a AIL method so-called AILBoost with policy boosting. In each round, the policy is ensembled considering the weakness of current policy. The ensemble training coefficient selection is simple with $\alpha (1-\alpha)$. It indicates that ensembling the policy is just to cover the small difference between expert and agent rollouts. Experiments show that AILBoost outperforms ValueDICE and DAC in state based and image based environments with small performance gains.

**Strengths:**

The method is easy to follow. Experiments are well conducted. The performance of AILBoost is convincing.

**Weaknesses:**

I do think this paper has some inspirations for AIL community, especially the experiment results are convincing.

Pls see Questions.

**Questions:**

I have a question about how the policies are ensembled? Is there a random number to choose which weak learner to perform? (this way keeps the multi-modalities for sampling) Or the policies are added with the outputs? Or any other ensemble methods? I am confused about this. However, ensembling the outputs of different policies may be wrong?

I do think it is a good idea to ensemble the weak learner for adversarial imitation learning. If the authors could resolve my confusion, I would like to raise my score.

---

> ### Author Response · Authors · 2023-11-20
>
> Thank you for your review. We hope to address your concerns below:
>
> __How policies are ensembled__
>
> The way we set up our ensemble is as follows: whenever we evaluate our policy, we choose policy $\pi_i$ from our ensemble with weight proportional to $\alpha_i$, and then roll out said policy. A more detailed discussion of this sampling process can be found in [1] section 4.1. As you pointed out, we sample the policy with respect to some random number, but note that our sampling is weighted based on the policy weight ($\alpha_i$). Note that we are learning a boosted policy in the state-action occupancy measure space as detailed in section 2 of [1]. Finally, the weighting scheme is following the gradient boosting literature for RL including [1][2].
>
> [1] Provably Efficient Maximum Entropy Exploration, Hazan et al. 2019
>
> [2] Local Policy Search in a Convex Space and Conservative Policy Iteration as Boosted Policy Search, Scherrer and Geist 2014

---

> > ### Author Response · Authors · 2023-11-23
> >
> > Hi,
> >
> > Hope you are doing well! We were wondering if you've had a chance to see our response regarding how we ensembled our policies. Please let us know if you have any additional questions. Thank you!
> >
> > Best,
> > Authors

---

> > > ### Comment · Reviewer_xvjX · 2023-11-23
> > > **Thanks for your response**
> > >
> > > Thanks for your response. I know this method can acquire the multi-modalities in the expert data. However, I think this ensemble method can do little to improve the final performance. Since each weak learner should be trained to be sub-optimal at least, I think the performance gain is limited. Therefore, I would like to keep my score.

---

### Official Review · Reviewer_Ah7S · 2023-10-28

**Soundness:** 3 good
**Presentation:** 3 good
**Contribution:** 3 good
**Rating:** 6
**Confidence:** 3

**Summary:**

This paper presents a novel algorithm called AILBoost for adversarial imitation learning (AIL) in the framework of boosting. AIL has been successful in various imitation learning applications, but existing methods have limitations in terms of off-policy training and sample efficiency. AILBoost addresses these limitations by maintaining an ensemble of weak learners (policies) and training a discriminator to maximize the discrepancy between the ensemble and the expert policy. A weighted replay buffer is used to represent the state-action distribution induced by the ensemble, allowing for training using the entire collected data. The algorithm is evaluated on state-based and pixel-based environments from the DeepMind Control Suite and outperforms existing methods in terms of sample efficiency and performance. The paper also discusses the benefits of boosting in AIL and the robustness of AILBoost across different training schedules.

**Strengths:**

Strengths of the Paper:

1. The paper introduces a novel algorithm, AILBoost, for adversarial imitation learning that leverages the framework of boosting. This approach is unique and different from existing methods in the field. The use of boosting in the context of AIL is a creative combination of ideas that brings new insights and potential benefits.

2. The paper provides a thorough and well-structured description of the AILBoost algorithm, including the theoretical foundations, algorithmic details, and empirical evaluation. The authors present a clear motivation for their approach and provide a comprehensive analysis of its performance compared to existing methods. The empirical evaluation is conducted on a benchmark suite of environments, and the results demonstrate the effectiveness and superiority of AILBoost.

3. The paper is well-written and easy to follow. The authors provide clear explanations of the concepts, algorithms, and experimental setup. The organization of the paper is logical, with sections dedicated to preliminaries, algorithm description, and experimental results. The use of figures and tables further enhances the clarity of the presentation.

4. The paper addresses important limitations in existing adversarial imitation learning methods, particularly in terms of off-policy training and sample efficiency. By introducing AILBoost, the authors provide a principled and effective solution that improves the performance and scalability of AIL algorithms. The empirical results demonstrate the significance of AILBoost, as it consistently outperforms existing methods across different environments and tasks.

**Weaknesses:**

Weaknesses of the Paper:

1. While the paper compares AILBoost with existing off-policy AIL algorithms such as DAC, ValueDICE, and IQ-Learn, it would be beneficial to include a comparison with more state-of-the-art IL methods as well. This would provide a more comprehensive evaluation and demonstrate how AILBoost performs relative to the best-performing IL algorithms in the field.

2. The paper lacks a detailed theoretical analysis of the AILBoost algorithm. While the authors provide some intuition and connections to boosting algorithms, a more rigorous theoretical analysis would strengthen the paper's claims. Specifically, providing formal proofs or guarantees of convergence, optimality, or sample complexity would enhance the theoretical foundation of AILBoost.

3. The paper briefly mentions the hyperparameters used in the experiments but does not provide a thorough discussion on their selection or sensitivity analysis. It would be valuable to explore the impact of different hyperparameter choices on the performance of AILBoost and provide insights into the robustness and generalizability of the algorithm.

4. The paper does not include ablation studies to analyze the individual components or design choices of AILBoost. By systematically removing or modifying specific components of the algorithm and evaluating their impact on performance, the authors could gain a deeper understanding of the contributions of each component and provide insights into their importance.

5. The paper briefly mentions that AILBoost may have increased memory cost due to maintaining weak learners. However, a more comprehensive discussion on the limitations of AILBoost, such as computational complexity, scalability to larger environments, or potential failure modes, would provide a more balanced perspective on the algorithm's practical applicability.

**Questions:**

1. Could you provide more insights into the computational complexity of AILBoost? Specifically, how does the algorithm scale with the number of weak learners, the size of the replay buffer, and the dimensionality of the state and action spaces? This information would be valuable for understanding the practical feasibility of AILBoost in larger and more complex environments.

2. In Section 4, you mention that AILBoost maintains a weighted replay buffer to represent the state-action distribution induced by the ensemble. Could you provide more details on how the weights are computed and updated in the replay buffer? Additionally, how does the size of the replay buffer affect the performance of AILBoost? It would be helpful to understand the trade-off between memory usage and performance.

3. The paper mentions that AILBoost achieves competitive performance with as little as one expert trajectory. Could you provide more insights into the limitations and trade-offs of using a small number of expert demonstrations? How does the performance of AILBoost change as the number of expert trajectories increases? It would be interesting to see if there is a threshold or diminishing returns in terms of performance improvement with more expert demonstrations.

4. In the experimental evaluation, you compare AILBoost with existing off-policy AIL algorithms. Could you provide more insights into the reasons behind the superior performance of AILBoost compared to these baselines? What are the key factors or design choices in AILBoost that contribute to its improved performance? This information would help in understanding the specific advantages of AILBoost over existing methods.

5. The paper briefly mentions the limitations of AILBoost, such as increased memory cost. Could you elaborate on other potential limitations or failure modes of the algorithm? Are there any specific scenarios or environments where AILBoost may not perform well? Providing a more comprehensive discussion on the limitations of AILBoost would help in understanding its practical applicability and potential areas for future improvement.

---

> ### Author Response · Authors · 2023-11-20
>
> Thank you for taking the time for a thorough review. We hope to address your concerns below:
>
> __[W1] More SOTA IL algorithms__
>
> We believe that we have covered a good range of SOTA off-policy, IL algorithms. Could the reviewer point us to any additional algorithms they would like to see?
>
> __[W2] Lack of Theoretical analysis__
>
> Thank you for this feedback. We agree with the reviewer that additional, in-depth analysis would only strengthen our work, but with this work we aimed to investigate and experimentally instantiate an algorithm that has the same principled off-policy treatment as ValueDICE and the practical scalability of DAC. Thus we focused on doing a thorough evaluation across the DeepMind Control Suite.
>
>
> __[W4] Lack of Ablation__
>
> Thank you for this critique. The practical instantiation of our algorithm is very close to DAC with the main difference being the boosted discriminator update. By comparing to DAC we felt that this was both ablating the most important contribution of AILBoost when compared to baselines. As for the ablation of the mixing parameter, we did not find this parameter to be so sensitive. Note that we used the same parameters across all experiments in a given task suite.
>
> __[Q1] Computational Cost__
>
> Of course, here is a discussion of the computation costs of our practical instantiation of Algorithm 1. We would like to clarify that in our practical instantiation of our algorithm, we incur negligible additional computational overhead over DAC. We maintain the same hyperparameters for the replay buffer and store the samples that we collected from each weak learner in the replay buffer of fixed size. The only additional data that we need to store are the weights of each sample that represent the weight of each weak learner. Finally, for both the policy and the discriminator, we found that warmstarting the following weak learner and discriminator with the previous models respectively worked well in practice. For the final evaluation, we evaluated both the ensemble and the final resulting policy and found both policies to have comparable performance. Thus, we could fully avoid storing the entire ensemble.
>
> __[Q2] How are the weights computed? What is the impact of replay buffer size?__
>
> For our experiments we implemented a fixed exponential weighting scheme that favors more recent weak learners. This is following gradient boosting work in RL [1] and maximum entropy RL [2]. We agree that it would be interesting to further investigate learning and adaptively update the ensemble weights during learning but did not find this to be essential in showing improvement over previous algorithms.
>
> One of the hyperparameters we explored involved the number of weak learners contained in the replay buffer. Given our implementation, increasing the buffer size would have the effect of increasing the number of weak learners combined in the ensemble. We did not find this hyperparameter to be very sensitive, especially since we could tune the weighting scheme to effectively control the number of relevant learners.
>
> [1] Local Policy Search in a Convex Space and Conservative Policy Iteration as Boosted Policy Search, Scherrer and Geist 2014
>
> [2] Provably Efficient Maximum Entropy Exploration, Hazan et al. 2019
>
> __[Q3] Effect of Number of Expert Demonstrations__
>
> AILBoost, similar to other IL algorithms, exhibits improved performance with more and more expert data. We would most likely see diminishing returns once we have enough expert demonstrations to fully estimate $d^{\pi*}$. This, however, is not specific to AILBoost and is a property of all AIL algorithms.
>
> __[Q4] What is the key factor?__
>
> Of course, first when compared to DAC, we believe the key improvement was AILBoost’s proper handling of off-policy samples in the discriminator update. ValueDICE also addresses and fixes this concern through the DICE objective; however, we have found this objective to be more unstable and difficult to scale in practice. Furthermore, the ValueDICE objective, in principle, only works for deterministic environments. Thus we believe the key factors are the combination of more scalable DAC-like optimization with an improved treatment of off-policy samples.
>
> __[Q5] Limitations__
>
> Thank you for providing this feedback. On top of the slight memory overhead, a potential limitation of AILBoost could be settings where SAC-style off-policy learning may be very expensive. That is, since SAC updates during the collection of a trajectory, in settings where it is impossible to incur the latency costs of stopping mid-trajectory to update the policy, AILBoost would also suffer.

---

### Official Review · Reviewer_o1rE · 2023-11-01

**Soundness:** 2 fair
**Presentation:** 3 good
**Contribution:** 2 fair
**Rating:** 6
**Confidence:** 3

**Summary:**

In this paper, the authors propose a novel adversarial imitation learning algorithm, AILBoost, which follows the framework of boosting. On the policy side, AILBoost maintains a weighted ensemble of policies and performs gradient boosting in the domain of state-action occupancy measures. On the discriminator side, AILBoost employs a weighted replay buffer combined with an expert dataset to train discriminators that maximize the discrepancy between the current mixed policy and the expert in an online manner. Empirically, AILBoost outperforms benchmarks in continuous control tasks across both controller state-based and pixel-based environments.

**Strengths:**

1.The author applies a mixed policy class which naturally connects with the buffer replay under the adversarial imitation learning framework, this is an interesting discovery.
2.By using a weighted mix of the learned policies, AILBoost performs gradient boosting, making the policy update smoother, which proves to be useful.

**Weaknesses:**

1.While thorough in its overall idea, this work lacks in-depth theoretical analysis. I suspect that noise during the policy rollout in each round might compound through the buffer replay, potentially disrupting the analysis of the discriminator.

2.The "weak learner" presented in this study doesn't conform to the traditional definition of weak learnability, which could be misleading. Contrary to AdaBoost, if the learner's base policy class fails to match the performance of the expert, then even mixing them at the initial state won't achieve expert-level performance.

**Questions:**

Questions:
1.Why using weighted mixing instead of using the DAgger[1] style average mixing?

2.Is the smooth loss condition in Section 4.1 hard to satisfy? Is the specific condition that $d^{\pi}$'s state action distribution covers the expert's state action distribution? Could you point out the exact theorem that makes this claim?

3.Can I get a definition for "off-policy"? I believe the proposed algorithm is still "on-policy" as it compares the current mixed policy state distribution to the expert's.

4.The noise during the policy rollout each round might compound via the buffer replay and disrupt the analysis on the discriminator's side. If this is the case and we need to gather new samples from the mixed policy every round, does this challenge the claim of being fully off-policy?

5. It would be interesting to see experiments where the learner is truly weaker than the expert, such that the expert is nonrealizable. Is weak learnability in imitation learning sufficient for near-expert performance?

6.In Algorithm 2, why do we calculate $\pi_{t+1}$ without the expert dataset? Is this practical? Any further justification?

7.Why in Algorithm 2, line 10, is the replay buffer given uniform weight for all samples? Any further justification?

8.Could you explain more about the meaning of 1000 policy updates per 100 discriminator updates in section 5.3?

9. As a follow up question from question 1, I believe the returned mixed policy is following the online reduction similar to DAgger[1], among DAgger style algorithms, MoBIL[2] is also using weighted policies, which may be worth mentioning.

Minor suggestions:

Please mention "10 trajectories" in the caption of Figure 3.
Please provide more description on the performance of behavior cloning in Figure 3.
Can you provide each task's plots corresponding to Figure 1 in the appendix? Also, all plots related to Figure 2 and Figure 3 would be appreciated.

[1]Ross S, Gordon G, Bagnell D. A reduction of imitation learning and structured prediction to no-regret online learning[C]//Proceedings of the fourteenth international conference on artificial intelligence and statistics. JMLR Workshop and Conference Proceedings, 2011: 627-635.
[2] Cheng C A, Yan X, Theodorou E, et al. Accelerating imitation learning with predictive models[C]//The 22nd International Conference on Artificial Intelligence and Statistics. PMLR, 2019: 3187-3196.

---

> ### Author Response · Authors · 2023-11-20
>
> Thank you for taking the time for a thorough review. We hope to address your concerns below:
>
> __[W1] Noise in Policy Rollout__
>
> Please see response/question to [Q4].
>
> __[W2] Not Aligned with Adaboost__
>
> We would first like to clarify that AILBoost is a gradient boosting [1] algorithm rather than one closer to adaboost. Moreover, could the reviewer elaborate on the unrealizable case? As [2] showed, our RL procedure could be viewed as boosted policy search.
>
> [1] Boosting Algorithms as Gradient Descent, Mason 1999
>
> [2] Local Policy Search in a Convex Space and Conservative Policy Iteration as Boosted Policy Search, Scherrer and Geist 2014
>
> __[Q1] Dagger mixing:__
>
> We do not consider the mixing performed in Dagger because we do not assume access to an online interactive expert. Instead, in our setting, we only have access to expert demonstration data. We are performing weighted mixing of weak learners, not a weighted mixture between a weak learner(s) and an expert, because we do not have access to an expert policy. More specifically, we are learning a boosted policy in the state-action occupancy measure space as detailed in section 2 of [1].
>
> [1] Provably Efficient Maximum Entropy Exploration, Hazan et al. 2019
>
> __[Q2] Smooth loss condition:__
>
> Please see [1], section 1.1 for their informal main theorem. For a more detailed discussion, please see Section 4 of [1].
>
> [1] Provably Efficient Maximum Entropy Exploration, Hazan et al. 2019
>
> __[Q3] Off-policy versus on-policy:__
>
> The underlying reinforcement learning algorithm that our boosting algorithm builds on top of is soft actor-critic (SAC). SAC is an off-policy algorithm because it keeps around a replay buffer of past experiences to update the policy (which weren’t collected from the current/most recent policy), vs. on-policy algorithms that strictly update the policy from samples gathered from the current policy. In our algorithm, we keep around a replay buffer of past weak learners’ experiences that we weight to update our mixture policy, making our algorithm off-policy.
>
> __[Q4] Noise during policy rollout:__
>
> Could we ask the reviewer to clarify what they mean by noise in the policy rollout? If they are referring to the case where the off-policy rollouts that we store are, by chance, that deviates from the experts then we emphasize that this is not a bad case. By training a discriminator that witnesses these deviations we provide a strong learning signal and over the course of training where we iterate discriminator training and policy sampling, we would eventually correct any distribution shift that may occur.
>
> __[Q5] Learner weaker than expert:__
>
> It would be interesting to see experiments where the learner is truly weaker than the expert. We will leave this investigation for future work.
>
> __[Q6] Updating the weaker leaner:__
>
> In algorithm 2, we update the weak learner $\pi_{t+1}$ with our off-policy RL algorithm using the reward function from the discriminator. The discriminator reward function captures the difference between the mixture policy and the expert data. The new weak learner is trained to minimize this difference, essentially making the mixture policy more similar to the expert dataset.
>
> __[Q7] Uniform weight weak learner update:__
>
> Our boosting algorithm modifies the discriminator updates to address the fact that the original DAC does not compute the density ratio – which is needed to minimize the original divergence that we care about. For updating our weak learner, we follow the original soft actor-critic (SAC) procedure which uses uniform weighting when sampling data from the replay buffer [1].
>
> [1] Soft Actor-Critic: Off-Policy Maximum Entropy Deep Reinforcement Learning with a Stochastic Actor, Haarnoja et al. 2018
>
> __[Q8] Policy updates versus discriminator updates:__
>
> By 1000 policy updates to 100 discriminator updates, we are referring to the gradient update steps that we did for each phase of learning. That is, we took 1000 SAC updates for the policy for every 100 gradient discriminator updates.
>
> __[Q9] Mobil reference__
>
> Thank you for the reference; we will make sure to include this MoBIL observation in our final version.

---

> > ### Comment · Reviewer_o1rE · 2023-11-23
> > **Quick question for the off-policy definition**
> >
> > Dear Authors,
> >
> > Thanks a lot for your explanation.  After reading ValueDICE paper and their definition of off-policy imitation learning, I am more confused if your algorithm is truly off-policy, since in line 3 of your Algorithm1 requires to collect on-policy samples. Isn't on-policy and off-policy describes the difference between using interactive environment to roll out learned policies or not?
> >
> > Could you please explain more with reference to justify the definition for off-policy imitation learning? Sorry for my late reply and I will follow up with other questions tonight.
> >
> > Best,
> > o1rE

---

> ### Author Response · Authors · 2023-11-23
> **Response regarding off-policy definition**
>
> Hi,
>
> Thank you so much for your response. The definition of "off-policy" in our case is that our algorithm uses data that is not collected by the most recent policy to update our ensemble. This is enforced through a replay buffer which we sample from to do the policy update. All data that is collected by our most recent policy is added to the buffer, and can be resampled for future updates. Thus, we are able to update our ensemble through data that wasn't explicitly collected by rolling out said ensemble, making our algorithm off-policy. An example of an off-policy method is Q-learning, where online interaction with the environment is allowed. In continuous control, SAC is an off-policy algorithm, which we used in our work.
>
> Offline reinforcement learning refers to when the agent cannot interact with the environment, which we think is what you are referring to. You are correct that our algorithm is not an offline RL/IL algorithm.
>
> Best,
> Authors

---

> > ### Comment · Reviewer_o1rE · 2023-11-23
> > **Remaining questions.**
> >
> > Dear authors,
> >
> > Thank you for your clarification, and I agree that the confusion regarding off-policy stems from preexisting definitions.
> >
> > The remaining questions and explanations are as follows:
> >
> > [W2] Thank you for explaining, and I agree that Adaboost belongs to gradient boosting, as does the proposed algorithm. Regarding the non-realizable case, consider a scenario like a combinational lock, where all policies in the class cannot match the state-action distribution of the expert, say there exists an epsilon performance gap for all policies. In this case, naively choosing one policy at random at the first state and executing it until the end will still result in the same epsilon performance gap. However, I agree that this mixed class is still stronger than the naive policy class.
> >
> > [Q1] No doubt, looking at the Dagger algorithm can be a bit confusing, since it involves mixing with the expert. However, the final guarantee for Dagger indeed pertains to the uniform mixture of the trained policy, which aligns with the approach of this work. Actually, the DAgger algorithm does not necessarily need to mix with the expert. It would be less confusing if you could review Proposition 2 of [1].
> >
> > [1] Li Y, Zhang C. On Efficient Online Imitation Learning via Classification[J]. Advances in Neural Information Processing Systems, 2022, 35: 32383-32397.
> >
> > [Q4] Regarding the noise in the policy rollouts, I mean that the state and action sampled from the distribution induced by rolling out the learner in the environment at each round cannot reflect the entire distribution, which induces a distribution-wise sample noise. This sample noise can possibly be learned by value functions and propagate to policies in future rounds.
> >
> > Based on your explanations, I am happy to raise my score to 6.
> >
> > Best, o1rE

---

### Official Review · Reviewer_mXvr · 2023-11-06

**Soundness:** 2 fair
**Presentation:** 2 fair
**Contribution:** 2 fair
**Rating:** 6
**Confidence:** 3

**Summary:**

Authors proposed AILBoost that exploits gradient boosting algorithm with the variational form of reverse KL divergence. Similar to the sample-based discrepancy minimization in AdaGAN (Tolstikhin et al., 2017), AILBoost adapted weighted replay buffer, where each trajectory and its importance are stored together in the replay buffer, which is later used to minimize the discrepancy between the expert's state-action occupancy measure and the mixture of agent's state-action occupancy measures. The empirical studies with controller-state-based and pixed-based environments show that AILBoost outperforms ValueDICE (Kostrikov et al., 2020) and DAC, both of which are well-known baselines in the AIL literature.

**Strengths:**

- Using gradient boosting with weighted replay buffer and applying it to adversarial imitation learning is interesting.
- Empirical results were evaluated with many metrics (IQM, Mean, Optimality GAP), which makes the results more reliable.
- Potential issues and limitations with the algorithm's scalability are discussed.
- Literature review on AIL is clearly done.

**Weaknesses:**

- Presentation should be improved: Abstract and Introduction (especially, Abstract) include too many details about existing works, which will be mentioned in Related Works and Preliminaries. Making Abstract and Introduction more succinct seems needed.
- The algorithm's complexity grows due to using all previous histories. However, this can be approximated by ignoring old samples, as mentioned by authors in Section 4.
- Contributions are focused on empirical sides and not on the theoretical sides; I don't think this is a crucial weakness, though.

**Questions:**

- More succinct Abstract and Introduction are needed. I think they are quite dense in its current form. One example I could think is adding a small figure to describe AILBoost's contribution, but this is not a mandatory comment to follow.
- There are two different forms of reverse KLD; (1) Donsker-Varadhan dual form from ValueDICE (2) Variational form from f-divergence GAIL. In AILBoost, the second form was used. Do we have any reason for not using the first form?
- The idea of using weighted replay buffer and boosting seems applicable to general AIL frameworks, although authors applied this only to the variational form of reverse KLD. Can we apply this idea to DAC, and if possible, can we see the performance for those cases? One reason I'm asking is the training procedure that combines DAC with boosting may be simpler than the one with rev KLD.
- In page 7, line 1, "we always warm start from $\pi_t$". ---> What does this mean? Also, Appendix A doesn't appear in the manuscript.
- AILBoost's computational complexity increases as $t$ increases in Algorithm 1. Can you please compare training times among DAC, ValueDICE and AILBoost?
- In Figure 1, "AILBoost outperforms DAC, ValueDICE, IQ-Learn, and BC across all metrics, amount of expert demonstrations, and tasks" ---> This is true from IQM perspective, but not true when 1 demonstration is considered with Mean and Optimality Gap.
- Figure 2,3,4's file sizes seem to be too large, which I think end up the heavy file size (34MB) of pdf file and makes text loading slow. Can we make those figures' sizes smaller?

---

> ### Author Response · Authors · 2023-11-20
>
> Thank you for taking the time for a thorough review. We hope to address your concerns below:
>
> __[Q1] Presentation of Abstract and Introduction__
>
> We would like to thank the reviewers for the feedback on the presentation of our paper. We agree that focusing on the paper’s main contribution and removing additional details about existing work could increase the readability of the abstract and introduction. We will address these concerns in the final version of the paper.
>
> __[Q2] Choice of reverse KL__
>
> DAC optimizes the same divergence that AIRL [1] minimizes, which is the reverse KL (please see Section 4.1 of [2]). Taking recommendations from both [2] and [3], we chose to minimize the functional form for the reverse KL for our adversarial IL setting. Finally, we independently confirmed that we can reproduce DAC’s results with this functional form of the reverse KL.
>
> [1] Learning Robust Rewards with Adversarial Inverse Reinforcement Learning, Fu et al. 2018
>
> [2] A Divergence Minimization Perspective on Imitation Learning Methods, Ghasemipour et al. 2019
>
> [3] Imitation Learning as f-Divergence Minimization, Ke et al. 2020
>
> __[Q3] Weighted replay buffer and boosting__
>
> We agree with the reviewer that our idea of incorporating a weighted replay buffer and boosting can be extended to any divergence minimization imitation learning algorithm. Note that DAC uses the reverse KL divergence in their algorithm, so we performed our experiments using this divergence for a fair comparison and thus we are improving DAC through boosting.
>
> __[Q4] Warm starting weak learner__
>
> When learning a weak learner in our boosting algorithm, we could either randomly initialize our weak learner or initialize our weak learner from our previous policy. On page 7, line 1, we are stating that our weak learners are initialized to the previous policy that was learned.
>
> __[Q5] Comparing training times__
>
> In terms of training time, the computational complexity remains the same – we still sample the same from the replay buffer for discriminator training (albeit weighted) and we only update the ensemble via SAC on the latest policy. The key difference is that we roll out with our ensemble, which doesn’t incur any computational overhead. We do make a point that there will be a slight increase in memory cost, which we found manageable at the end of Section 4.
>
> __[Q7] Figure Sizes__
>
> Thank you for pointing out the issues with figure 2,3,4. We will make sure to address these issues for the final version.

---

### Meta-Review · Area_Chair_QBKn · 2023-12-08

**Metareview:**

This paper develops a boosting-based approach to off-policy adversarial imitation learning. The approach uses a weighted ensemble of policies to minimize state-action discrepancy with the expect. Performance improvement are demonstrated over existing SOTA methods. This paper makes a nice contribution by leveraging a classical idea (boosting) to address the practical challenges of off-policy adversarial imitation. The authors position the contributions of the paper well with respect to existing work and clearly describe the approach. The main weakness is the lack of theoretical analysis/guarantees (despite strong empirical results), including some statement explaining the ability of combining weak learners (policies) to achieve strong performance. Overall, the paper is borderline in terms of ratings, but the positives outweigh the negatives and I recommend acceptance.

**Justification For Why Not Higher Score:**

Better theoretical analysis/guarantees would be needed to improve the score.

**Justification For Why Not Lower Score:**

The paper makes a nice contribution using a method that is explained clearly and positioned nicely with respect to related SOTA methods, which it outperforms empirically.

---

### Decision · Program_Chairs · 2024-01-16

Accept (poster)